# Comparative Analysis on Machine Learning and Deep Learning to Predict Post-Induction Hypotension

**DOI:** 10.3390/s20164575

**Published:** 2020-08-14

**Authors:** Jihyun Lee, Jiyoung Woo, Ah Reum Kang, Young-Seob Jeong, Woohyun Jung, Misoon Lee, Sang Hyun Kim

**Affiliations:** 1SCH Media Labs, Soonchunhyang University, Asan 31538, Korea; jihyun@sch.ac.kr (J.L.); armk@arkang.net (A.R.K.); bytecell@sch.ac.kr (Y.-S.J.); 2Department of Anesthesiology and Pain Medicine, Soonchunhyang University Bucheon Hospital, Soonchunhyang University College of Medicine, Bucheon 14584, Korea; mulungtomato@naver.com (W.J.); misoonlee@schmc.ac.kr (M.L.)

**Keywords:** hypotension prediction, machine learning, deep learning, anesthesia, vital records, biomedical sensor

## Abstract

Hypotensive events in the initial stage of anesthesia can cause serious complications in the patients after surgery, which could be fatal. In this study, we intended to predict hypotension after tracheal intubation using machine learning and deep learning techniques after intubation one minute in advance. Meta learning models, such as random forest, extreme gradient boosting (Xgboost), and deep learning models, especially the convolutional neural network (CNN) model and the deep neural network (DNN), were trained to predict hypotension occurring between tracheal intubation and incision, using data from four minutes to one minute before tracheal intubation. Vital records and electronic health records (EHR) for 282 of 319 patients who underwent laparoscopic cholecystectomy from October 2018 to July 2019 were collected. Among the 282 patients, 151 developed post-induction hypotension. Our experiments had two scenarios: using raw vital records and feature engineering on vital records. The experiments on raw data showed that CNN had the best accuracy of 72.63%, followed by random forest (70.32%) and Xgboost (64.6%). The experiments on feature engineering showed that random forest combined with feature selection had the best accuracy of 74.89%, while CNN had a lower accuracy of 68.95% than that of the experiment on raw data. Our study is an extension of previous studies to detect hypotension before intubation with a one-minute advance. To improve accuracy, we built a model using state-of-art algorithms. We found that CNN had a good performance, but that random forest had a better performance when combined with feature selection. In addition, we found that the examination period (data period) is also important.

## 1. Introduction

Hypotension is a hemodynamic abnormality commonly observed during anesthesia. Intraoperative hypotension directly affects postoperative mortality and morbidity [1,2,3,4,5,6,7]. Undoubtedly, careful monitoring, early prediction of hypotension, and immediate response are necessary to improve the survival rate and reduce life-threatening complications after surgery. However, predicting hemodynamic changes are not easy, despite the careful efforts of anesthesiologists and the use of modern monitoring equipment. Thus, developing a system that can overcome these clinical limitations can serve as an alternative diagnostic tool. If such a system that can predict the occurrence of hypotension at bedside is available, it may lead to reduced postoperative complications and mortality. Machine learning is currently applied to various clinical medical areas and several studies related to the prediction of hemodynamic changes using artificial neural networks have recently been published [8,9,10]. To develop this prediction model, it is essential to collect, store, and immediately analyze large data that arise from patient medical records and anesthesia. Currently, most hospitals have electronic health records (her), order communication systems, and photo archiving and communication systems. With network-based medical records, more data can be stored and analyzed than ever before. In addition, vital records generated by various monitoring devices can be extracted, stored, and integrated in real time, to be used for the development of prediction models [11]. Considering this, the development of a hypotension prediction model during anesthesia using machine learning is expected to improve the prognosis after surgery, as well as provide a useful alternative tool for anesthesiologists who require rapid and accurate decisions and prompt responses. In this study, we intended to develop a deep learning model to predict hypotension in advance using intraoperative vital records collected during anesthesia and extracted from EHR.

### Related Work

Previous studies focused on identifying predictors of hypotension. Ko et al. [12] revealed that lactate levels, blood in the nasogastric tube, and systolic blood pressure values are risk factors for the occurrence of hypotension in patients with non-arrhythmic upper gastrointestinal bleeding. Lee et al. [13] evaluated shock index (SI), modified shock index (MSI), and age shock index (SI) as predictors of hypotension after tracheal intubation, using a machine learning technique and logistic regression. Reich et al. [14] tested the association of hypotension, gender, age, and height for 10 min after anesthesia induction, preoperative drug therapy, and anesthesia induction, using a Fisher’s exact test. Südfeld et al. [15] investigated the association between post-induction hypotension (PIH; i.e., arterial hypotension occurring during the first 20 min after anaesthesia induction) and early intraoperative hypotension (eIOH; i.e., arterial hypotension during the first 30 min of surgery). Nonaka et al. [16] revealed the predictors of hypotension after carotid stent through the Mann–Whitney U test and the Kruskal–Wallis test, and introduced a scoring system for predicting hypotension.

Advanced research has attempted to build a prediction model based on the predictors from previous works. We reviewed related studies on hypotension prediction using a machine learning algorithm, focusing on adopted features and algorithms and compared the performance of key studies. Ghosh et al. [17] predicted acute hypotensive episodes with a sequential contrast mining algorithm using data extracted from the MIMIC-II critical care research database. Janghorbani et al. [18] compared the two models after constructing a logistic regression model and support vector machine model for predicting acute hypotensive episodes using time series data, including HR (heart rate), SAP (systolic arterial pressure), DAP (diastolic arterial pressure), and MAP (mean arterial pressure). Park et al. [19] attempted a study to predict intradialytic hypotension in patients undergoing hemodialysis, by combining the basic and heart rate variability (HRV) variables. The basic variables in this paper are diabetes mellitus (DM), coronary artery disease (CAD), cause of ESRD, ultrafiltration rate, and vintage of dialysis. Moghadam et al. [20] constructed an algorithm to predict hypotension in the intensive care, by obtaining the mean and standard deviation of 11 variables, including arterial blood pressure (ABP), heart rate (HR), systolic blood pressure (SBP), diastolic blood pressure (DBP), respiration rate (Resp), peripheral capillary oxygen saturation (SpO2), pulse pressure (PP), MAP, cardiac output (CO), MAP-to-HR ratio (MAP2HR), and average of respiratory rate (RR) intervals on electrocardiogram (ECG) time series (RR).

Kendale et al. [8] have attempted multiple machine learning analyses, using features extracted from EHR data to predict hypotension that occurred 10 min post-induction. The features for machine learning were selected using the recursive feature function, and the prediction model using the gradient boosting machine showed the strongest performance in both the training set and the test set. In our study, the period from the induction of anesthesia to the incision was divided into two sections, and the data before intubation were used for training and the data after intubation was used for testing.

Lin et al. [21] used age, gender, weight, height, hematocrit, American Society of Anesthesiologists (ASA) score, basal SBP, basal DBP, basal HR, history of hypertension, history of diabetes, surgical category, emergency, and dose of local anesthetics as features, and built a prediction model of hypotension episodes during spinal anesthesia, using artificial neural networks and logistic regression algorithms. In addition to these features, we also used anesthetic concentration, bispectral index, and mechanical ventilation data in three-second increments.

Hatib et al. [9] extracted the variables for each period by dividing the arterial blood pressure waveform into five sections: systolic phase, diastolic phase, systolic rise phase, systolic decay phase, and overall decay phase. The algorithm developed by them showed excellent performance by showing high sensitivity and specificity 5, 10, and even 15 min before the occurrence of hypotension.

However, most of these studies on the development of a prediction model of hemodynamic change using machine learning are based on EHR that have been collected in a traditional way or data obtained from a small number of patient monitoring devices. There are invasive and non-invasive methods for measuring blood pressure during surgery. The invasive method to measure arterial blood pressure and data is continuously recorded. Non-invasive methods are discontinuous, and are usually measured every 5, 2, or 1 min. Continuous data is advantageous for predicting blood pressure with high accuracy, but arterial blood pressure measurement is recommended only for high-risk patients because it can be a burden for patients. In other words, using invasive measurements results in better predictions, but is not feasible in most patients.

Furthermore, previous works built models to predict the right blood pressure 15 or 10 min after induction using machine leaning. None of the studies focused on hypotension during intubation.

We summarized related studies in terms of taxonomy composed of outcome, outcome type, features, and algorithms and compared the performances of key papers in Table 1. We categorized previous studies into static and dynamic models. The static model makes predictions on hypotension at a certain point in time, but the dynamic model makes predictions over the entire period with a certain delay.

The static model with the best performance achieved 77.6% [21]. Our previous work adopted vital records and delayed the prediction time at intubation, to use vital records as long as possible. This achieved 80% accuracy, which improved the work of Lin et al. [13]. Here, the accuracy refers to the ratio of data that is properly classified among all data, and AUC (area under the curve) refers to the area under the roc curve drawn with recall (sensitivity) and 1-specificity. Sensitivity refers to the ratio of true positives over the sum of true positives and false negatives. Specificity refers to the ratio of true negatives over the sum of true negatives and false positives. Accuracy and roc curve are frequently used as a model performance evaluation scale. In this work, we aimed to extend the prediction time point before intubation, to give medical staff time to prepare.

## 2. Materials and Methods

### 2.1. Data Collection

Adult patients (age ≥18 years) who underwent laparoscopic cholecystectomy under general anesthesia at Soonchunhyang University Bucheon Hospital, Bucheon City, Republic of Korea between 30 October 2018 and 31 July 2019 were included in this retrospective study. Vital records were collected from a Bx50 monitor, BIS monitor, Orchestra (Anesthetic pharmaceutical injection machine), and Datex-Ohmeda (Anesthesia machine-artificial breathing), using a vital recorder, and EHR data were retrieved from the database. The construction of the database was approved by our institutional review board (approval No. 2018-06-012). Additional approval from our institutional review board was obtained for this study (approval No. 2020-05-039). A snapshot of the vital recorder (https://vitaldb.net) is displayed in Figure 1. This vital recorder collects vital records from various devices and tracks patient status. Figure 1 visualizes the data of all parameters from various devices that collected during surgery. We selected the parameters related to the occurrence of hypotension among these data, and used them for training and testing to develop a hypotension predictive model. Lee et al. [22] predicted the bispectral index by constructing a long short-term memory and the feed-forward neural network model, using data collected from vital recorders. The entire dataset used in our experiment is listed in Table 2.

We set the time interval at 3 s to unify the vital-sign record, and replaced it with the last value if there was a gap in the data due to the time difference. Among the 319 patients, 282 patients were tested, excluding those whose the time of tracheal intubation was missing and whose the time gap between anesthesia induction and tracheal intubation were less than 4 min. Since data from 4 min to 1 min before tracheal intubation are used as input data, the data with a period of less than 4 min are excluded. The longest data point time of the input data was 15 min 33 s (311 s) and the shortest data point time was 1 min 54 s (38 s). Data with outliers were replaced with the average of the precedent and antecedent values. For each parameter, a measurable range was specified, and cases exceeding the range were manually checked. In rare cases, there are records of defects or error values of measuring equipment. For example, when the concentration of the propofol was recorded as 538,976.25, it was replaced with the previous concentration value of 3.467. The characteristics of the patients extracted from the database are shown in Table 3. Baseline blood pressure was defined as the initial blood pressure in the operating room. The type of surgery is laparoscopic cholecystectomy, the longest data point time is 15′33 (311 s), and the shortest data point time is 1′54 (38 s).

Propofol and remifentanil were injected to induce loss of consciousness, after which rocuronium was administered intravenously to facilitate tracheal intubation. Thereafter, anesthesia was maintained with the target propofol and remifentanil concentrations titrated by the attending anesthesiologist’s clinical judgment, based on the patient’s vital records.

Vital recorders display electrocardiography, pulse oximetry, intermittent noninvasive blood pressure measurements, and bispectral index scoring. General anesthesia was induced and maintained with total intravenous anesthesia (TIVA) using propofol and remifentanil via a target-controlled infusion (TCI) pump (Orchestra Base Primea with module DPS; Fresenius Kabi AG, Germany).

### 2.2. Features

The features used in this study were 27 including plethysmogram oxygen saturation, end-tidal CO2 partial pressure, propofol, and remifentanil collected from a vital recorder, as shown in Table 3. Propofol and remifentanil are intravenous anesthetics and include plasma concentration, effect-site concentration, and target concentration. The demographic variables collected by EHRs included age, sex, height, weight, BMI, and ASA score, and 29 comorbidities including cardiovascular disease and respiratory disease. The frequency and duration of hypotension between 4 min before and during tracheal intubation and vasopressor and vasodilator usage were used. Table 2 lists the features collected from the vital recorder and EHR. We investigated the features that had significant differences between hypotension and non-hypotension groups. Beforehand, Shapiro–Wilk tests for normality were performed. Depending on the normality, the continuous variables were subjected to the t-test if they were normally distributed, and Wilcoxon rank-sum test if they were not normally distributed and the categorical variable was subjected to the chi-squared or Fisher’s exact test. The results shown in Table 3 indicate that vital records are more significant than EMR, and that all BIS and drug information have significant differences between hypotension and non-hypotension.

### 2.3. Algorithms

We adopted random forest, extreme gradient boosting (Xgboost), deep neural network (DNN) and convolutional neural network (CNN) models for predicting hypotension. The adopted algorithms are briefly introduced as follows. Random forest, first introduced in 2001 by Breiman [23], is an ensemble machine learning model used for classification and regression analysis. It is a representative algorithm that uses bagging among ensemble techniques. Multiple decision trees are formed, new data points are passed through each tree at the same time, and the most voted result is selected as the final classification result by voting on the results classified by each tree. By creating a large number of trees, the effect of overfitting is reduced. The biggest difference from the existing ensemble model is that randomness is applied to variable features as well as observation.

Xgboost is a typical example of a boosting technique among ensemble techniques. It is an algorithm that adds a regularization term to solve the overfitting problem, which is a problem of gradient boosting (GBM). Boosting is an algorithm that improves prediction or classification performance by combining multiple weak learners that are sequential [24]. Similar to bagging, a number of classifiers are made from the initial sample data, but unlike bagging, the previous model works by sequentially weighting the data that is not predicted, so that the next model can learn more. Xgboost is a machine learning algorithm that is frequently used because it uses a parallel processing technique to speed learning and classification, and to prevent overfitting.

In this study, we adopted two different types of algorithms from the deep learning category. The deep learning algorithm is one of the advanced techniques of machine learning and it achieves a deeper understanding of the input data without human interpretation. Deep neural networks build successive layers to extract meaningful representations from data in learning patterns. DNN is a feedforward neural network with two or more hidden layers. Chen et al. [25] built a DNN model and showed good results in predicting intradialysis hypotension in hemodialysis patients. The CNN introduced by Lecun [26] is an artificial neural network that captures local patterns, and is mainly used for processing voice and image data. In traditional machine learning, the algorithm performs on features extracted from data, while the deep learning algorithm extracts features from raw data and automatically identifies patterns of these features. The CNN is composed of the convolution layer, activation function, and pooling layer. Here, the convolution layer is a process of extracting the features of the data, and the adjacent components of each component are examined to identify the characteristics. The activation function is a function that receives a signal, processes, and outputs it. The pooling layer refers to the process of reducing the size of the layer by extracting the ‘representative value’ from the values of the previous layer after going through the convolution layer. The effect of the pooling layer is a reduction in the data dimension and a reduction in the risk of overfitting. The types of pooling include max pooling and average pooling. In the medical field, CNN was developed to classify arrhythmias into 12 classes, and produced good results [27]. In their study, inputs included vasoactive drug, hypotension information, vital records, and EHR data between tracheal intubation, 4 min before tracheal intubation. The CNN model was built with the vital records data, and the EHR data were built using the fully connected layer, and then the two models were combined using concatenate.

### 2.4. Problem Definition and Labeling Approach

In this study, we aimed to build a model for predicting hypotension after tracheal intubation. We split the vital record into two periods: the first period is from the start of anesthesia to tracheal intubation, and the second period is from tracheal intubation to incision. Models were constructed using vital records, demographic data, and comorbidities data in EHR during the first period, excluding the data during the delay time. The delay time was set at 1 min for preparation against hypotension. The delay time implies that targets will be tracked after 1 min.

The hypotension was tracked before surgery and after intubation, regardless of its frequency. We defined a patient as someone who experienced hypotension at least once. The class was extracted as 1 if hypotension was present, and 0 if otherwise. Hypotension was defined as SBP < 90 mmHg or mean blood pressure (MBP) < 65 mmHg. The lookback period was set from the 1-min data just before tracheal intubation to the data for the preceding 3 min. Figure 2 shows how training data and their classes are set.

After building the model, we can predict hypotension from tracheal intubation until the onset of surgery, using data obtained from tracheal intubation during anesthesia induction in a new patient.

### 2.5. Model Building

Our experiments are twofold. First, deep learning and machine learning models were built using raw data. For comparison, a predictive model is created using deep learning and machine learning models using statistical features extracted from raw data. For feature extraction, we adopt simple statistics, including min, max, mean, and standard deviation, to extract abnormal patterns. To determine the best combination of features, we perform the feature selection as well and compare the performance. Figure 3 shows the framework of developing the post-induction hypotension prediction model with comparative analysis of traditional machine learning and deep learning. We explore three datasets of raw vital records, EHR and vasoactive drug for deep learning. For deep learning, we test two models of CNN and DNN.

For CNN, although the data are different for each patient, the input was sliced into 60 data points and each point was 3 s, reflecting the lookback period, which was 3 min in Figure 2. The input is an image-like matrix with a size of (60, 27, 1), where 60 refers to the length of time steps and 27 are the features.In the CNN, to find the hyperparameters, the autokeras, a keras package built in R to find the optimal model. The optimal model consists of 2 convolution layers and a 1 max pooling layer, as shown in Figure 4. The convolution layer consists of several convolution filters that selectively extract the input value. The pooling layer taking the max of the values wrapped by the filter. For the hidden layers, we adopt a ReLU function as an activation function. After reducing the dimension by convolution layer and max pooling, it goes through a flattening process to convert it into one dimension and send it to the fully connected layer. In the fully connected layer, the activation function is set to a sigmoid function. The RMSProp is used as the optimizer to make the learning speed fast and stable. In addition, a drop-out layer is used to avoid overfitting.

The experiment will be performed three folds (1) vital records (27 features), (2) vital records and EHR (56 features), and (3) vital records, EHR, and vasoactive drug (67 features). They were performed through the Keras merge operation using the layer_concatenate function as a multi-input model. Build a CNN model with vital records and EHR data, build a model using a fully connected layer, then combine the two models using concatenate, and then add a sigmoid classifier. The second experiment used vital records statistics data. Figure 4 shows the architecture of the CNN model, which consists of two convolution layers, followed by the max pooling layer and the drop-out layer, and finally dense layer is used for classification. Table 4 shows the specifications on parameters of the CNN architecture model in Figure 4.

The overall architecture of the DNN model is shown in Figure 5. The deep neural network consists of four fully connected layers. “Fully connected” refers to the fact that all neurons between two adjacent layers are fully pairwise connected. The vital record should be flattened to be feed to the DNN model. The values of vital feature at the time step are regarded as variables independently. The last layer is formed as a sigmoid layer configured to classify hypotension as (0, 1) after tracheal intubation. The optimizer in DNN was set to ADAM optimization algorithm. In DNN, the batch normalization and drop out layers are used to avoid overfitting. The loss function represents how the network measures performance in training data and optimizes it to reduce model output errors to correctly classify hypotension after tracheal intubation. The loss function for binary classification uses the binary_cross entropy.

For traditional machine learning, all values of every three seconds are too many to be used as features. Thus, we developed a summary for each period of data using statistical features. The 27 input vital records variables for 3 min were summarized using min, max, mean, and standard deviation statistics. There are a total of 97 features, including derived features, and the experiment was conducted a total of 4 times through the feature selection process. (1) All derived features (98 features) (2) feature set 1 (remove redundant features; 45 features) (3) feature set 2 (rank features by importance; 20 features) (4) features set 3 (recursive feature elimination; 29 features). We employed three feature selections for dimension reduction: remove redundant features, rank features by importance, and recursive feature elimination [28]. First, removing redundant features is a way of removing redundant features from a dataset. Data can contain attributes that are highly correlated with each other. Machine learning algorithms tend to perform better if highly correlated attributes are removed. In our experiment, features with a correlation coefficient of 0.5 or more were removed, and experiments were conducted with 45 of the 97 features.

We constructed a random forest model, and estimated the importance of the features, and then tested it with 20 features of high importance among the 97 features. The final method was recursive feature elimination. Automatic feature selection methods can be used to build many models with different subsets of a dataset, and identify those attributes that are or are not required to build an accurate model. This automatic feature selection method is called recursive feature elimination, or RFE. In this study, the RFE method was used to test 30 out of 97 features. Table A1 in Appendix A shows the variables used in each feature set. Finally, after selecting the best performance model in each experiment, the results were compared for 2 min and 1 min before the prediction time point. The model with the best results in the first experiment was the vital records and EHR (56 features), and the best performance model in the second experiment was the feature set 2 model tested with random forest.

### 2.6. Performance Evaluation

We performed repeated k-fold cross-validation to guarantee unbiased performance. The k-fold cross-validation method is a statistical skill to measure the performance of the model on new data, after splitting the data into k folds. A fold is tested as a new data for the model built from the remaining k-1 folds, and this process is repeated, while all folds are tested once. The k-fold validation has randomness in sample selection in forming a fold. When the samples are homogeneous, the randomness would not cause biased performance on a specific fold split. However, when samples are heterogeneous, the algorithm performance could change, depending on which samples are split into which fold. The repeated k-fold validation complements this weakness by repeating the step splitting samples into folds n times. Bio-medical data, especially our vital records, are diverse depending on the patient, so we repeated 10-fold cross-validation 100 times to generate a stable performance. We also calculated precision, recall, and accuracy to evaluate the performance of the model [29]. Accuracy is calculated as (TP + TN)/(TP + FN + FP + TN), precision is calculated as TP/(TP + FP) and recall is calculated as TP/(TP + FN), where TP, TN, FP, and FN represent true positives, true negatives, false positives, and false negatives, respectively. We displayed the precision and recall for only hypotension class, because two classes of hypotension and non-hypotension are well balanced so the good performance of one class, and the accuracy implies the good performance of other classes. For a highly imbalanced case for a certain class, the accuracy metric can be biased to a major class. In this case, AUC is appropriate, because the curve balances the class sizes by adjusting the threshold to judge a sample for a binary classification.

## 3. Results

After the exclusion of cases without tracheal intubation time and patients who took more than 4 min from anesthetic induction to tracheal intubation, of all 282 patients, 151 (53.55%) had hypotension. The hypotension group were older in age (56.5 ± 14.5 years) and had a lower body weight (24.7 ± 3.6 kg). In the first experiment, in the model for predicting the occurrence of hypotension using CNN, the performance was the best when the vital records data and EHR data were used as features, as shown in Table 5.

In the CNN model, EHR improves the accuracy by as much as 0.39% (from 72.24 to 72.63). However, the vasoactive drug information is not effective in detecting hypotension; it even lowers the accuracy. All algorithms exhibit better performance in recall than precision. This indicates when these algorithms are adopted in the operating room, double-checking procedure that filters low-risk cases from the cases detected, as hypotension may be required to improve the precision. In traditional machine learning, RF that is an assemble tree composed of the several feature subsets on several data subsets is comparative to CNN. RF and CNN have a common characteristic in that they characterize the subset of features instead of the whole feature set. Rather than characterizing the entire feature set, it is more effective to characterize the feature subset in a repetitive way.

The second experiment was to build a model using the statistical features of vital records. As shown in Table 6, the random forest model generated the best performance on statistical features, with 74.89%. Of the two machine learning models, the random forest performed better than Xgboost, especially with feature set 2. Overall, feature selection except the remove redundant features method improves the performance across all algorithms. This experiment shows that deep learning algorithms work poorer with statistical features rather than raw features. Statistical features such as mean, standard deviation, min and max definitely extract abnormal patterns from hypotension and the random forest, an ensemble tree, can build the best performance model. CNN conceives the local pattern from the feature set, but it turns out the conceiving of local pattern on statistical features is not effective in distinguishing abnormal pattern. For example, the convolution on min and max values might neutralize the abnormal patterns. Statistical features are better to be dealt independently. Thus, compared to CNN, the random forest outperformed. Interestingly, Xgboost exhibits poor performance on both cases, we infer that the sequential update is not effective because all cases are diverse from each other, so the current update for several cases does not guarantee a better performance for upcoming cases.

Finally, we performed an experiment by varying the lookback period with the best models in the previous experiment. The best model in the first experiment was 2) vital records and EHR (56 features), and the best model in the second experiment was feature set 2. We reduced the lookback period from 3 min to 1 min. The results of predicting hypotension with 3 min of data were the best than predicting hypotension, with data of 2 min and 1 min in Table 7. The vital records during surgery are continuous time series data, and it is advantageous to predict the trend of blood pressure, as the look back period is longer. If the hypotension can be predicted even with a short lookback period, the size of input data will be smaller, and it will be more useful for clinical use.

Feature importance was determined for the final model, if this is a possibility for the given machine learning algorithm, as not all machine learning algorithms are amenable to computing variable importance. Feature importance is computed based on how important any given feature is to aid in the classification process when the classifier is built, determined by its effect on the performance measure [30]. Generally, feature importance helps to assess the impact of any given variable on the performance of the algorithm. If a variable with high importance is permuted or removed from the model, the performance decreases. The greater the importance, the more essential the variable is to the performance of the model. Figure 6 shows the importance of the statistical features. Overall, the min and mean of NIBP_SBP and NIBP_MBP have high importance in terms of the accuracy and the gini index.

## 4. Discussion

Our previous work set the problem to the classification of patients who will experience hypotension or not after intubation. The previous work set up the prediction time at the point of intubation. This previous work used 126 patients. To extract information from vital records, we derived features using simple statistics, and focused on the method of the feature selection to apply machine learning algorithms. The previous work mainly utilized traditional machine learning, such as the naïve Bayes, logistic regression, RF, and ANN (artificial neural network, a simple form of DNN). To improve the proposed model to be implemented in medical rooms, we needed a prediction model that can give alarm in one-minute advance before intubation. The advanced alarm gives a time to medical staff to prepare for hypotension. For example, the preparation will help reduce the level of anesthetic drug. Furthermore, the feature selection method is critical to improve the performance in the previous work. This required extra time as much as training. Thus, we proposed the prediction model that enables prediction in advance and are free of extensive feature engineering, including feature construction and selection.

In this study, we investigated whether a machine learning algorithm can predict the occurrence of hypotension during anesthesia in advance. We trained our prediction model using vital records generated, between 4 min and 1 min before tracheal intubation. Among the predicted models tested, the random forest, an ensemble tree, showed the best performance using the statistical features on vital records, and CNN model showed the second-best performance using raw vital records. Our previous model using RF showed superior performance compared to our current model. The reason might be that the current study obtained data from 4 to 1 min before tracheal intubation.

From the experiments in our current work, the random forest model using only 20 features had the best accuracy, precision, and recall than the model, using all the 97 features. We found that simple statistical features such as mean, standard deviation, min and max are enough to extract abnormal patterns from hypotension. In the algorithm perspective, the random forest selects the subset of features and the subset of samples, builds the various models and builds ensemble trees. Xgboost exhibits poor performance on both cases. From that, we concluded that the vital records are diverse for patients, so a unified model is not appropriate to be deployed across all patients. The ensemble approach with limited important features is good to build hypotension prediction model. In this study, we explored possibilities to apply deep learning to vital records in hypotension prediction. We found that simple statistical features associated with ensemble trees are slightly more effective than the deep learning algorithm. However, RF requires statistical feature extraction, which is not required for CNN. Furthermore, the statistical features need the feature selection process. The experiments showed that the method of “Rank features by importance” improves the performance. This feature selection method is computationally expensive, because it checks the importance of every feature. To adopt our proposed models in real-world operating rooms where patients are diverse in their vital and their vital is required to be used to update the pre-built model, computationally heavy algorithm is not appropriate. In real-world operation rooms where real-time detection is required, RF and CNN should be employed complementarily. RF can be used after batch update, and CNN can be used without pre-processing and batch update.

In current work, we evaluated our models with the small number of patients. In our future work, we will check the feasibility of the random forest model for large number of patients. We have previously described the development of a prediction model using an ANN [10]. Our previous model is comparable to Kendale et al. [8], and achieves a better result in a similar condition. Most of these studies on the development of a prediction model of hemodynamic change using machine learning are based on EHR, that have been collected in a traditional way or data obtained from a small number of patient monitoring devices. The data analyzed included high-resolution time-synchronized physiological and pharmacological data from various anesthetic devices during surgery. In particular, data analyzed in our study came from mechanical ventilators and anesthetic workstation that have been overlooked in most literature in the past. In recent years, research on prediction models using machine learning has been steadily published in several clinical fields, such as arrhythmia, postoperative mortality, morbidity, and hypotension [10,21,31,32,33,34]. This trend clearly leads to many advantages, such as the development of the medical environment, improved patient safety, and improved prognosis. A recently published literature about prediction models using machine learning can be developed, as a useful tool to prevent various adverse events that can seriously affect a patient’s prognosis [35]. Jeong et al. [36] developed a real-time model for predicting 3-min-ahead blood pressure during the anesthesia induction phase. They collected vital sign data from various devices, such as patient monitors, anesthesia machines and infusion pumps. They showed mean absolute errors between 8.2 and 11.1 mmHg using a bidirectional recurrent neural network. The results of their model are unsuitable for clinical use, but this is a new attempt to predict blood pressure in real time, using a deep learning model, and if the performance of this model improves in the future, it can be used as an alternative diagnostic tool in the operating room.

This study has several limitations. First, features used for prediction model training were extracted only from the intraoperative data from 4 to 1 min before tracheal intubation. This is a relatively short period, and it is inevitable that the amount of collected vital record and features are small. We aimed to develop a model to predict the occurrence of hypotension after tracheal intubation using data from/anesthetic induction to tracheal intubation. While CNN trained the extracted features in a relatively short time, we demonstrated the feasibility of developing prediction models of hypotension. Second, our study included only patients who underwent laparoscopic cholecystectomy. The reason for this is that, because the anesthesia procedure for the same surgery and the drugs used are similar, the effect on the intraoperative data can be easily controlled.

## Figures and Tables

**Figure 1 sensors-20-04575-f001:**
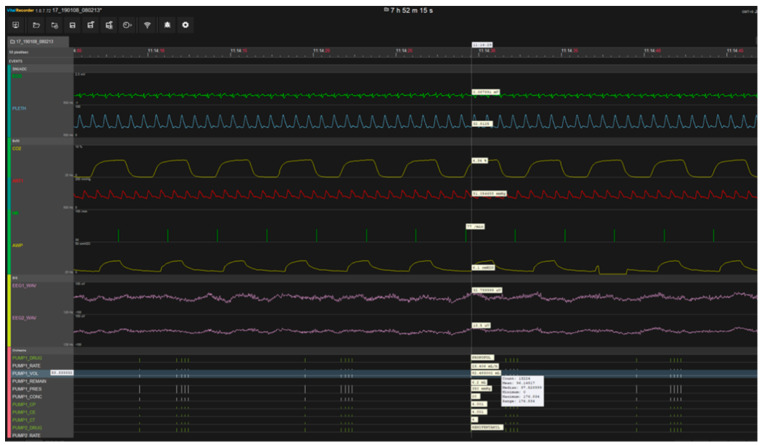
Vital Recorder.

**Figure 2 sensors-20-04575-f002:**
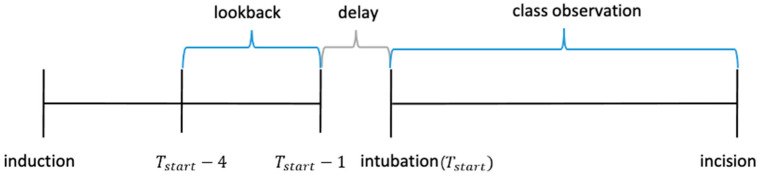
Preparing training data.

**Figure 3 sensors-20-04575-f003:**
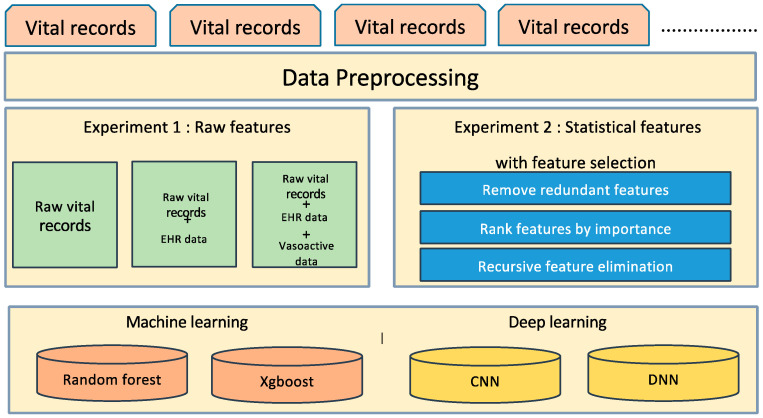
Research framework of hypotension prediction composed of the deep learning model and the machine learning model, using raw features and statistical features.

**Figure 4 sensors-20-04575-f004:**
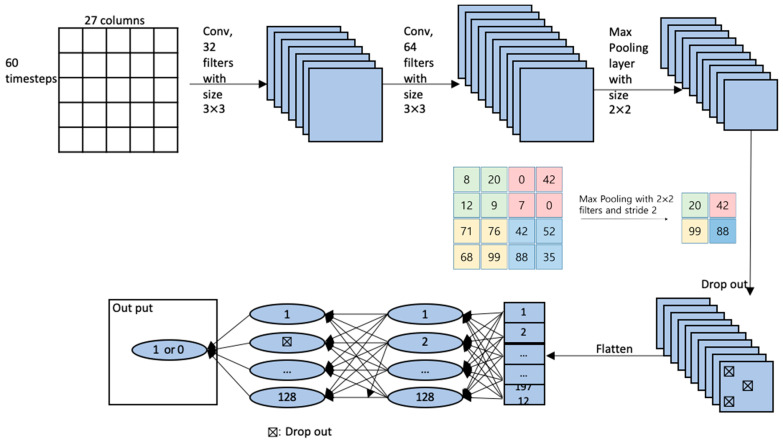
CNN architecture and vital record processing for CNN.

**Figure 5 sensors-20-04575-f005:**
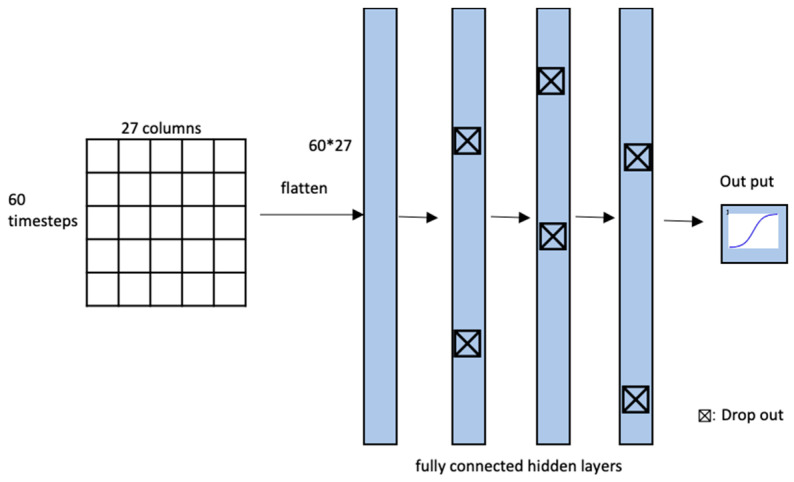
DNN architecture and vital record processing for DNN.

**Figure 6 sensors-20-04575-f006:**
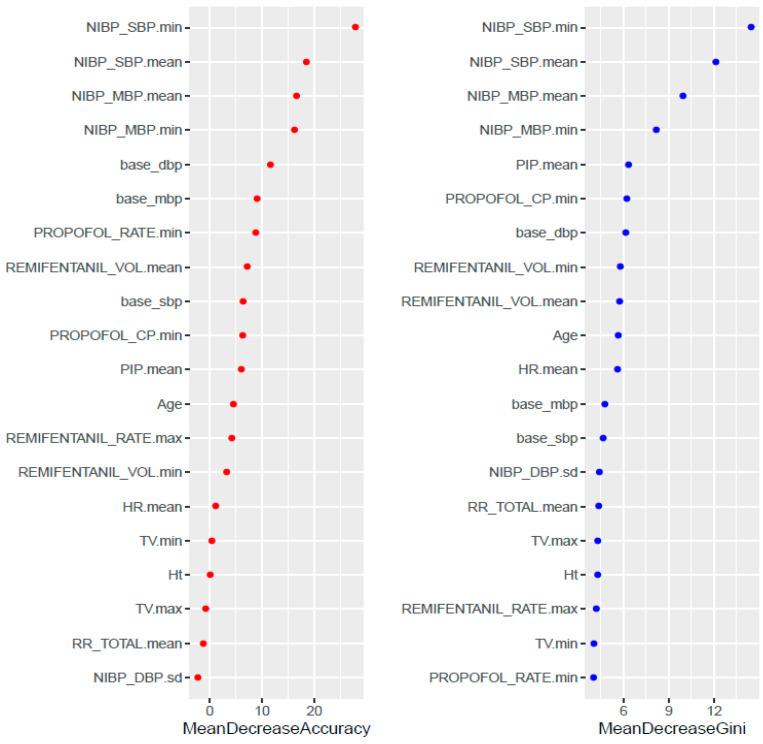
Feature importance.

**Table 1 sensors-20-04575-t001:** The taxonomy of related studies on hypotension prediction.

Study	Outcome	Outcome Type	Feature	Algorithm	Performance
Park et al. [19]	Hypotension before 1 month after surgery for hemodialysis patients	Static, after surgery	Heart rate variability (DM, CAD, CHF, Age, UFR, iPTH, ARB or ACEI, CCB, b-blocker, Mean HR, RRI, SDNN, RMSSD, VLF, LF, HF, TP, LF/HF ratio)	Multivariate negative binomial model	AUC: 0.804
Moghadam et al. [20]	At least 5 min before hypotension	Dynamic	ABP (arterial blood pressure), HR, Sys, Dia, Resp, SpO_2_, PP, MAP, CO, MAP to HR ratio (MAP2HR), average of RR intervals on ECG time series (RR)	Logistic Regression (LR)	Accuracy: 95% Sensitivity: 85% Specificity: 96%
Kendale et al. [8]	Hypotension within 10 min after induction	Static, at induction	Age, Sex, BMI, ASA Score, Medical comorbidities, Preoperative medication, Intraoperative medications, Mean peak inspiratory pressure, First mean arterial pressure, Time of day, non-invasive and invasive blood pressure	LR, Support Vector Machine, Naïve Bayes, K-Nearest Neighbor, Linear Discriminant Analysis, Random Forest, Neural Network, Gradient Boosting Algorithm	Sensitivity: 64% Specificity: 75% AUC: 0.76
Lin et al. [21]	Hypotension within 15 min after induction for spinal anesthesia	Static, at induction	Age, Gender, Weight, Height, Hematocrit, ASA score, Basal SBP, Basal DBP, Basal HR, History of hypertension, History of diabetes, Surgical category, Emergency, Dose of local anesthetics	LR, ANN, Simplified ANN	Accuracy: 77.6% Sensitivity: 75.9% Specificity: 76%
Hatib et al. [9]	Hypotension at least within 5 min	Dynamic	3022 features from arterial pressure waveform: Signal features, floTrac features, COTrek features, complexity features, Baroeflex features, variability features, spectral features, Delta change features	LR	Sensitivity: 86.8% Specificity: 88.5% AUC: 0.95

RRI: R-R interval, SDNN: standard deviation of N-N interval, RMSSD: squared root of the mean squared differences of successive N-N interval, VLF: Very low frequency, LF: low frequency, HF: high frequency, TP: total power, Sys: systolic blood pressure, Dia: diastolic blood pressure, Resp: respiration rate, PP: pulse pressure, CO: cardiac output, MAP2HR: Map-to-HR ratio.

**Table 2 sensors-20-04575-t002:** Data description.

Data Source	Categories	Features
Electronic Health Record	Demographic data	Age Sex Height Weight Body mass index ASA classification Base Systolic Blood Pressure Base Diastolic Blood Pressure Base Mean Blood Pressure
	Comorbidities	Cardiovascular disease
		Respiratory disease
		Gastrointestinal disease
		Renal disease
		Endocrine disease
		Neurologic disease
	Baseline	Systolic
		Mean
		Diastolic
Vital Recorder	Noninvasive blood pressure	Systolic
		Mean
		Diastolic
	Heart rate	Heart rate
	Mechanical ventilation data	Plethysmogram oxygen saturation
		End-tidal CO2 partial pressure
		NMT_TOF_CNT
		Respiratory rate
		Tidal volume
		Minute ventilation
		Peak inspiratory pressure
		Positive end expiratory pressure
	Bispectral index	Spectral edge frequency
		Signal quality index
		Electromyogram power
		Total power
		Bispectral index value
	Anesthetic drug	Rate
		Plasma concentration
		Effect-site concentration
		Target concentration
		Volume
	Vasoactive drug administration	Vasopressor
		Vasodilator
	Hypotension	Frequency
		Duration
		Average duration

**Table 3 sensors-20-04575-t003:** Patient characteristics.

Characteristic	All Patients (n = 82)	Hypotension (n = 151)	Non Hypotension (n = 131)	*p*-Value
Age	54.7 (14.1)	56.5 (14.5)	52.6 (13.4)	0.023 *
Sex (male)	134 (47.5%)	66 (43.7%)	68 (51.9%)	0.209
Height	162.1 (9)	161.2 (8.7)	163.2 (9.2)	0.067
Weight	66.6 (12.4)	64.5 (12)	69.2 (12.5)	0.002 **
BMI	25.2 (3.6)	24.7 (3.6)	25.8 (3.5)	0.019 *
ASA classification -no				0.426
1	95 (33.7%)	48 (31.8%)	47 (35.9%)	
2	151 (54.6%)	82 (54.3%)	72 (55%)	
3	33 (11.7%)	21 (13.9%)	12 (9.1%)	
Comorbidities				
Cardiovascular disease				
Hypertension	97 (34.4%)	55 (36.4%)	42 (32.1%)	0.520
Atrial fibrillation	2 (0.7%)	2 (1.3%)	0	0.500
Coronary artery disease	5 (1.8%)	4 (2.6%)	1 (0.8%)	0.377
Angina pectoris	5 (1.8%)	2 (1.3%)	3 (2.3%)	0.666
Congestive heart failure	1 (0.4%)	1 (0.7%)	1 (0.8%)	1.000
Valvular heart disease	1 (0.4%)	1 (0.7%)	0	1.000
Respiratory disease				
Asthma	17 (6%)	14 (9.3%)	3 (2.3%)	0.027*
Chronic obstructive pulmonary disease	6 (2.1%)	3 (2%)	3 (2.3%)	1.000
Gastrointestinal disease				
Hepatitis	3 (1.1%)	2 (1.3%)	1 (0.8%)	1.000
Liver cirrhosis	6 (2.1%)	3 (2%)	3 (2.3%)	1.000
Viral carrier	6 (2.1%)	3 (2%)	3 (2.3%)	1.000
Hepatitis B viral infection	12 (4.3%)	5 (3.3%)	7 (5.3%)	0.584
Hepatitis C viral infection	2 (0.7%)	1 (0.7%)	1 (0.8%)	1.000
Renal disease				
Chronic kidney injury				0.209
2	1 (0.4%)	0	1 (0.8%)	
3	3 (1.1%)	3 (2%)	0	
4	1 (0.4%)	1 (0.7%)	0	
End-stage renal disease	1 (0.4%)	1 (0.7%)	0	1.000
Endocrine disease				
Diabetes mellitus	62 (22%)	37 (24.5%)	25 (19.1%)	0.341
Thyroid disease				0.667
1	3 (1.1%)	2 (1.3%)	1 (0.8%)	
2	4 (1.4%)	1 (0.7%)	3 (2.3%)	
3	8 (2.8%)	5 (3.3%)	3 (2.3%)	
Neurologic disease				
Cerebrovascular disease	12 (4.3%)	8 (5.350	4 (3.1%)	0.525
Cerebral aneurysm	1 (0.4%)	0	1 (0.8%)	0.465
Baseline blood pressure -mmHg				
Systolic	146.2 (23.8)	140.8 (23.1)	152.3 (23.1)	0.001 ***
Mean	105.9 (16)	102.4 (16.2)	109.9 (15)	0.001 ***
Diastolic	80.9 (10.5)	78.3 (10.5)	83.9 (9.8)	0.001 ***
Noninvasive blood pressure -mmHg				
Systolic	113.3 (22.7)	105.4 (18.1)	122.3 (24)	<0.001 ***
Mean	84.3 (15.5)	78.9 (12.5)	90.4 (16.2)	<0.001 ***
Diastolic	66.6 (12.5)	62.7 (10.8)	71.2 (12.8)	<0.001 ***
Heart rate -/min	70.6 (13.1)	70.8 (12.9)	70.3 (13.3)	<0.001
Mechanical ventilation data				
Plethysmogram oxygen saturation	99.4 (1.5)	99.4 (1.7)	99.4 (1.5)	<0.001 ***
End-tidal CO2 partial pressure -%	2.5 (1.5)	2.5 (1.5)	2.4 (1.5)	0.001 ***
NMT_TOF_CNT	2 (1.9)	2.1 (1.9)	1.9 (1.9)	0.001 ***
Respiratory rate -/min	15.7 (8.5)	15.8 (8.6)	15.5 (8.4)	0.210
Tidal volume -mL	242.4 (172.3)	242.3 (168.8)	242.5 (176.2)	0.318
Minute ventilation -L/min	4.2 (2.7)	4.2 (2.7)	4.2 (2.8)	0.169
Peak inspiratory pressure -cmH2O	16.5 (7.4)	16.1 (7)	16.9 (7.7)	<0.001 ***
Positive end expiratory pressure -cmH2O	3.1 (2.3)	3.1 (2.2)	3.1 (2.3)	0.480
Bispectral index				
Spectral edge frequency -Hz	17.1 (3.7)	17.1 (3.7)	17 (3.7)	0.001 ***
Signal quality index -Hz	87.4 (16.2)	88.6 (15.3)	86.1 (17.2)	<0.001 ***
Electromyogram power -Hz	30.5 (6.8)	30.1 (6.4)	30.9 (7.3)	0.001 ***
Total power	63 (7.5)	63 (7.5)	63.1 (7.4)	0.007**
Bispectral index value	51.8 (16.6)	52.2 (16)	51.5 (17.3)	0.001 ***
Anesthetic drug				
Rate				
propofol -mg	52.6 (91.4)	47.6 (60.2)	58.4 (117.2)	<0.001 ***
remifentanil -mg	8.3 (33.6)	7.2 (30.2)	9.5 (37.2)	0.001 ***
Plasma concentration				
propofol -mg	5.3 (2.2)	5.2 (2)	5.4 (2.4)	<0.001 ***
remifentanil -mg	2.1 (1.7)	2.1 (1.6)	2.2 (1.8)	0.001 ***
Effect-site concentration				
propofol -mg	4.9 (1.1)	4.9 (0.9)	4.9 (1.2)	<0.001 ***
remifentanil -mg	1.7 (0.9)	1.6 (0.9)	1.7 (0.9)	<0.001 ***
Target concentration				
propofol -mg	4.9 (1)	4.9 (0.9)	4.9 (1.1)	<0.001 ***
remifentanil -mg	1.6 (1)	1.5 (1)	1.6 (1.1)	0.001 ***
Volume				
propofol -mg	6.7 (2.5)	6.4 (4.9)	7 (3)	<0.001 ***
remifentanil -mg	0.8 (0.7)	0.7 (0.4)	0.9 (1)	<0.001 ***
Vasoactive drug administration -no				
Ephedrine	4 (1.4%)	4 (2.6%)	0	0.126
Phenylephrine	1 (0.4%)	1 (0.7%)	0	1.000
Nicardipine	1 (0.4%)	0	1 (0.8%)	0.465
Esmolol	1 (0.4%)	0	1 (0.8%)	0.465

*p* < 0.001 ***, *p* < 0.01 **, *p* < 0.05 *.

**Table 4 sensors-20-04575-t004:** Specifications of parameters for CNN.

Layer Type	Input Shape	Filter Shape	Activation	Parameters, #
Input	(60, 27, 1)			0
Conv2D	(58, 25, 32)	(3, 3)	Relu	320
Conv2D	(56, 23, 64)	(3, 3)	Relu	18,496
MaxPooling2D	(28, 11, 64)	(2, 2)		0
Dropout	(28, 11, 64)			0
Flatten	(19712)			0
dense	(128)		Relu	2,523,264
Dropout	(128)			0
dense	(1)		sigmoid	129

**Table 5 sensors-20-04575-t005:** Results on prediction performance using raw features.

Feature Set	Performance Metrics		Random Forest	Xgboost	CNN	DNN
Vital records	Accuracy		70.32	64.15	72.24	63.25
	Hypotension	Precision	69.97	65.92	72.1	64.2
		Recall	78.28	69.15	79.04	72.12
Vital records + EHR	Accuracy		70.26	64.32	72.63	63.4
	Hypotension	Precision	69.84	66.14	72.69	64.38
		Recall	78.37	68.99	79.33	71.99
Vital records + EHR + Vasoactive drug	Accuracy		70.28	64.6	71.87	63.22
	Hypotension	Precision	69.82	66.5	72.92	64.32
		Recall	78.35	69.05	76.37	71.95

**Table 6 sensors-20-04575-t006:** Results on prediction performance using statistical features.

		Random Forest	Xgboost	CNN	DNN
All features (97)	Accuracy	70.76	65.15	65.33	69.03
	Precision	72.16	67.37	68.29	70.78
	Recall	74.72	68.61	68.54	72.79
Feature set 1	Accuracy	65.26	61.75	60.34	63.03
	Precision	67.02	64.81	63.79	65.57
	Recall	70.88	63.93	66.76	67.22
Feature set 2	Accuracy	74.89	69.84	67.95	73.85
	Precision	75.8	71.5	70.69	73.72
	Recall	78.43	73.17	71.78	79.93
Feature set 3	Accuracy	73.06	68.28	68.95	73.84
	Precision	74.59	70.19	74.11	75.73
	Recall	75.97	71.35	66.97	75.88

**Table 7 sensors-20-04575-t007:** Results on prediction performance with different lookback periods.

		3 Min	2 Min	1 Min
Vital records + HER with CNN	Accuracy	72.63	70.37	70.39
	Precision	72.69	71.06	71.53
	Recall	79.33	75.64	74.64
Feature set 2 with RF	Accuracy	74.89	71.45	74.42
	Precision	75.8	72.16	72.66
	Recall	78.43	76.26	75.17

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
