# Peer review of "Comparative Analysis on Machine Learning and Deep Learning to Predict Post-Induction Hypotension"

_sensors, 2020, doi:10.3390/s20164575_

Round 1

Reviewer 1 Report

Thanks for an interesting paper comparing the performance of different machine and deep learning algorithms to predict hypotension. I have some maior and minor points to consider for improving the manuscript:

Maior:
Abstract: It should be made clear that the numbers 72.63% and so on are accurancies.
Line 96-102: This section is very confusing. I think I understand that the intend is to argue that using invasive measurements results in better predictions but is not feasible in most patients, but you text is very unclear, and could be interpreted as arguing the opposite.
Line 111-114: The comparion of related studies is confusing as you do not comment on the difference between accurancy and AUC, and how this should be interpreted. In general it would be good to make more clear, why accurency is your main outcome measure. I do not necessarily disagree with using accurance, but it should be explained more.
Line 138-140: The explanation why only 282 of 319 patients could be included is unclear.
Line 140-141: Please specify how outliers where identified.
Table 2: This table is superfluous, and the four numbers could just be mentioned in the text.
Line 177-213: This whole section is confusing, as it is unclear what you did in this study, and what you refer from other studies. As this section is the main part of your study it should be much clearer.
Section 3 and Table 6: You should discuss the (small) differences between some of the algorithms, instead of solely focusing on which results in the highest accurancy.
Line 350: It is unclear which models "Best models" refer to, this should be added (possibly to Table 8).
Line 405-409: This section is unclear.
Discussion: You should comment on differences between the algorithms with respect too their possible application in practice.

Minor:
Line 67: Fisher+s exact test with capital "F" (Fisher was a person)
Line 74. The "etc." is confusing.
Table 1: Please give the meaning of the abbreviations in the "Algorithm" column, e.g. "LR".
Line 142: Do you mean "Table 3" instead of "Table 2"?
Line 160: What does the number "27" indicate?
Line 170: I presume you mean "Wilcoxon" and not "Wilcox" and you shouuld specify if you used the rank-sum or sign-rank test.
Line 171: "Chi" should be with lowercase "c", as it is not a name.
Table 4: Please report all p-values with 3 decimal digits, it is confusing to read that some of them are with fewer og more digits. Furthermore, consider if ">0.99" should be replaced by "1.000", as the ">0.99" makes it harder to read.
Line 240: The fragment starting with "Comparision" is not a sentence.
Line 305: "K-fold" should be with lowercase "k"
Line 327: You should be consistent in using "biomedical signal" or "vital records".
Table 6: Why is "72.24" in bold?
Line 342-343: Then sentence starting with "Thus" does not make sense.
Line 373: You have a "the" too much.

Author Response

Please find attached a revised version of our manuscript “Comparative analysis on machine learning and deep learning to predict post-induction hypotension”, which we would like to resubmit for publication as a research article in Sensors.

Your comments were highly insightful and enabled us to greatly improve the quality of our manuscript. In the following pages are our point-by-point responses your comments.

We hope that the revisions in the manuscript and our accompanying responses will be sufficient to make our manuscript suitable for publication in Sensors.

We shall look forward to hearing from you at your earliest convenience.

Reviewer 2 Report

The authors developed a deep learning model to predict hypotension in advance using intraoperative bio-signal data collected during anesthesia and extracted from EHR. The authors compared random forest, extreme gradient boosting, deep neural network (I guess standard feed-forward network with more than one layer), and convolutional neural network in developing their prediction model. The authors first preprocessed the raw EHR and vital records data to determine the features that will be used for the prediction model. Then, depending on the algorithms used, the authors either further processed the data or just used the raw data as is.

For the first two models (random forest and Xgboost), the authors first extracted the usual statistics from the raw data and built the prediction model from the statistics of the data. For the deep learning models, the authors used the raw data as is. The author then discussed the results of the comparison. Overall, the authors provided a nice and useful comparison between some of the most used learning methods for hypotension prediction.

I list below my suggestions and comments for the authors:

  1. In Section 1.1, although the summaries of existing literature are provided, it is hard to see how the literatures are related to the work of the authors. In my opinion, after summarizing the literatures, the reader might understand better if the authors also relate how their works improve/complement existing results.
  2. It is not clear what the reader is supposed to see from Figure 1. Are all the plots in Figure 1 used in the training? 
  3. In Section 2.3., in the first paragraph, the authors mentioned that they used random forest, Xgboost, and CNN to train the model. However in the second paragraph, the authors mentioned that they used two different deep learning algorithms: deep neural network and CNN. Looking at the Results section, it seems to me that the authors ended up using 4 methods instead 3 methods.
  4. Still in Section 2.3, the authors mentioned that they used deep neural networks and CNN. However from my understanding, CNN is an algorithm is in the class of deep neural network algorithms. Looking at Section 2.5, it seems to me that what the authors mean by DNN is the Feed-Forward Neural Network with hidden layers. Is my understanding correct?
  5. It is not clear what I am supposed to see in Figures 3 -- 5. Please consider better captions and explanations for the figures.
  6. In Section 3, the authors discussed three different experiments: model building from raw data, model building from statistical extract, and performance with different lookback value. In both model building experiments, the authors built the models using all the algorithms. However, my understanding from Section 2.5 is that raw data is used only to build model by deep learning algorithms while statistical extract is used only to build model by the standard machine learning algorithms.
  7. The experiment on lookback period gives an expected result. I don’t see the significance of this part.

Author Response

(The authors gave the same response as above.)

Round 2

Reviewer 1 Report

Thank you for the changes, the manuscript has much improved. I have no further comments, apart from a recommendation of checking the English language one additional time, as there still are some gramatical errors, typos and clumsy phrases.

Author Response

Thank you for your help. We double checked typos and errors.